# Exploring spatial clusters of caesarean sections across India - Insights from National Family Health Survey data

Durgesh Prasad Sahoo[1], Biju Soman[2], Neeraj Agarwal[1], Shreyas Patil[1], Adrija Roy[3], Arun Mitra[1]*

**1** Department of Community Medicine and Family Medicine, All India Institute of Medical Sciences (AIIMS), Bibinagar, Hyderabad, India, **2** Sree Chitra Tirunal Institute for Medical Sciences and Technology (SCTIMST), Trivandrum, India, **3** RVM Institute of Medical Sciences and Research, Siddipet, Telangana, India

☯ These authors contributed equally to this work.

\* dr.arunmitra@gmail.com

## Abstract

The rising prevalence of caesarean sections (C-sections) in India, surpassing the World Health Organization's recommended rate of 10–15%, has become a significant public health concern. This study aims to analyze the patterns and spatial clustering of C-sections across India's states and districts, utilizing data from the National Family Health Survey (NFHS-4 and NFHS-5). Spatial autocorrelation analysis was applied to summarize the extent to which districts with similar proportions of C-sections tend to occur next to each other, forming spatial clusters. This was done using Global Moran's I index for the fourth and fifth rounds of the National Family Health Survey (NFHS) data conducted in 2015–16 and 2019–21, respectively. LISA cluster maps were produced to test for global spatial autocorrelation, and the Getis-Ord General G statistic was used to test for local clustering. Our research reveals a substantial increase in caesarean deliveries, from 17.2% in NFHS-4 (2015–16) to 21.5% in NFHS-5 (2019–21). This trend is more pronounced in rural areas, challenging previous assumptions about urbanization as a primary driver. We observed marked disparities between public and private healthcare facilities, with private hospitals showing significantly higher rates of caesarean births. Public hospitals exhibited clustering in southern India and parts of northern India, while private facilities showed clustering along the eastern coast and in northeastern regions. Global Moran's I and Getis-Ord General G statistics confirmed significant spatial autocorrelation and clustering (p<0.01) for both public and private facilities. This study highlights the complex interplay of factors influencing caesarean section rates, including healthcare access, institutional policies, and the urban-rural differential. It underscores the need for targeted interventions and policy adjustments to address the disproportionate increase in caesarean deliveries.

**Data availability statement:** The data used in the current study are available in the open domain. Moreover, all code and data analysis workflows, including the spatial files are available at a public repository (https://github.com/drarunmitra/nfhs-caesarean-paper-data-analysis) along with the computational workflow in R. This is also available at https://doi.org/10.5281/zenodo.17789152.

**Funding:** The author(s) received no specific funding for this work.

**Competing interests:** The authors have declared that no competing interests exist.

## Introduction

Cesarean delivery, commonly referred to as a C-section, is a surgical intervention involving incisions in the maternal abdomen and uterus to facilitate childbirth. This procedure is generally indicated when vaginal delivery presents potential risks to the mother or the fetus, or when complications occur during labor [1]. Maternal mortality and morbidity are more prevalent following cesarean delivery as compared to normal vaginal delivery, with an increased risk of complications such as uterine rupture, abnormal placentation, ectopic pregnancy, stillbirth, and preterm birth. Furthermore, emerging evidence suggests that infants delivered via cesarean section are exposed to distinct hormonal, physical, bacterial, and medical environments, which may subtly influence neonatal physiology [2]. The World Health Organization (WHO) recommends that the rate of cesarean deliveries be maintained between 10% and 15% (10–15 cesarean deliveries per 100 live births) to ensure optimal outcomes for both mothers and newborns [3]. The global rate of cesarean deliveries has risen steadily, currently accounting for over 21% of all childbirths. Various studies indicates that this trend is likely to persist, with nearly 29% of all births expected to occur via cesarean section by 2030 [4]. Similarly, in India, the prevalence of cesarean deliveries rose from 17.2% in 2016 to 21.5% in 2021 [5].

The persistent increase in cesarean delivery rates in low- and middle-income countries (LMICs), including India, has raised significant concerns and has been the focus of many researchers [6]. Another concern is that the likelihood of undergoing a cesarean delivery in a private facility is more than double that in a government facility [7]. The rising trend in cesarean deliveries has been associated with a variety of factors. One key influence is the evolving maternal and fetal indications for performing C-sections, which may reflect advancements in medical understanding and technology. Additionally, changes in maternal preferences, driven by factors such as perceived safety and convenience, have contributed to this rise. Furthermore, shifts in healthcare policies and practices, including the availability of resources in both public and private sectors, play a significant role in determining the rates of cesarean sections [8–10]. Evidence suggests that the rate of cesarean deliveries worldwide reflects significant disparities both between and within countries [6]. In India, a similar pattern of geographical variation is observed, with significant regional disparities in C-section rates, ranging from 5.2% in Nagaland to 60.7% in Telangana [11]. Previous studies report disparities in C-section rates across wealth quintiles in the country [12,13]. Similar observations have reported in studies from other lower and middle-income countries (LMICs) as well, where the C-section rates were the lowest among the rural poor and highest among the urban wealthy [14,15]

Previous studies show the growing disparities in access to quality maternal healthcare, as well as inequities in the distribution of these services across different geographical regions and sociocultural contexts [5,11,16,17]. Recent studies have also begun to employ spatial analytical techniques to examine C-section patterns at the subnational level. While some researchers investigated C-section delivery patterns in South India using Bayesian multilevel modelling, others applied Getis-Ord Gi* hotspot analysis and multiscale geographically weighted regression to examine

spatial clustering and determinants in northeastern India [18,19]. However, these studies focus on specific subregions and rely on a single survey round (NFHS-5). A comprehensive, pan-India spatial analysis examining the temporal evolution of C-section clustering across both NFHS-4 and NFHS-5, with facility-type disaggregation for public and private health sectors, remains a critical gap. The present study addresses this gap by providing the most comprehensive district-level spatial analysis of C-section clustering across India, examining trends across two survey rounds and stratifying by facility type — generating evidence to inform targeted public health policy and health systems strengthening.

### Objectives

1. To analyze the trends and patterns of caesarean section deliveries at both the state and district levels in India

2. To examine the spatial distribution and clustering of caesarean section rates across districts in India

## Methodology

### Data source

The NFHS-5 (2019–2021) is the fifth iteration of the National Family Health Survey (NFHS), which is a large-scale, multi-round survey that provides essential data on health and family welfare for policy and programme development in India. The NFHS-5 has been conducted in two phases (Phase I: 17 June 2019–30 January 2020; Phase II: 2 January 2020–30 April 2021) and has been designed to provide district level estimates for a wide range of health indicators by collecting data from a representative sample of 636,699 households including 724,115 women, and 101,839 men [20].

The data from the NFHS are provided as fact sheets (individual *.PDF files) for states, union territories and districts, which can be downloaded from the NFHS website (http://rchiips.org/nfhs/index.shtml). Similar district-level fact sheets for NFHS-4 (2015–16) are also available for access [21]. The national level estimates presented in this study were directly obtained from the national level fact sheets and are therefore expressed as 'proportions' and 'differences in proportions'. Zone level estimates were derived by aggregating data from the state-level fact sheets and are reported as 'mean proportions' and 'mean differences in proportions'. The state-level estimates for urban vs. rural comparisons were directly derived from the state level fact sheets and are reported as 'proportions' and 'differences in proportions.

### Data extraction & Preparation

The data extraction process was performed using the statistical programming language R (version 4.4.2) in three distinct stages [22]. The first stage involved downloading the district level fact sheets (*.pdf documents) and systematically organising them into sub-folders. This was achieved through web scraping techniques (utilizing the rvest and fs packages) in R, to extract the district-level fact sheets from both NFHS-5 and NFHS-4 datasets and structure them into an organized file hierarchy [23,24].

The second stage involved extracting the relevant variables from the fact sheets. Seven key indicators related to delivery care (for births occurring in the five years prior to the survey) were identified and extracted using the tabulapdf package which is based on the Tabula java library for computationally extracting tables from PDF documents [25]. The data extracted from the pdf documents were then saved as spread sheets for verification and manual corrections were applied where necessary. Subsequently, the individual spreadsheets were merged to construct the district-level dataset for both NFHS-4 and NFHS-5. Similarly, state level fact sheets were processed to generate the state-level datasets [26].

The third and final stage involved integrating the state and district-level datasets with their respective shape files (*.shp), which contain administrative boundary data. These shapefiles were sourced from the online map's portal of Survey of India, Government of India (2021) [27]. Due to a discrepancy between the number of districts in the NFHS dataset (704 districts) and the number of districts in the shapefiles provided by the Survey of India (727 districts), missing data

were imputed using the relevant state-level averages. The tidyr and dplyr packages were used to tidy, manipulate, and merge the datasets into the required formats (See Fig 1). The preprocessed shape files were imported into R version 4.4.2 for further analysis and visualization [28,29]. Additional details of the methods used and a complete list of R packages used is provided in the supplement (S1 Text).

## Ethics statement

Due ethical processes were followed and ethical clearance was obtained from the Institute Ethics Committee vide no. AIIMS/BBN/IEC/AUG/2024 dated 26.08.2024

## Statistical analysis

**Descriptive statistics.** National, state, and district-level estimates were represented as 'proportions' and 'differences in proportions', directly extracted from their respective fact sheets. All zonal level estimates were represented as "mean proportions" and 'mean differences in proportions' as they were derived from the state-level fact sheets. District-level maps for overall c-section births as well as c-section births in public and private health facilities, were generated by linking the calculated proportion of c-sections in each district to the corresponding shape files obtained from the Survey of India [27]. Analysis of variance (ANOVA) was performed to test the difference in proportion of c-sections births across place of residence, place of birth and NFHS survey rounds. A p-value of $<0.05$ was considered statistically significant throughout the analysis. All statistical analysis was performed using R version 4.4.2 [22].

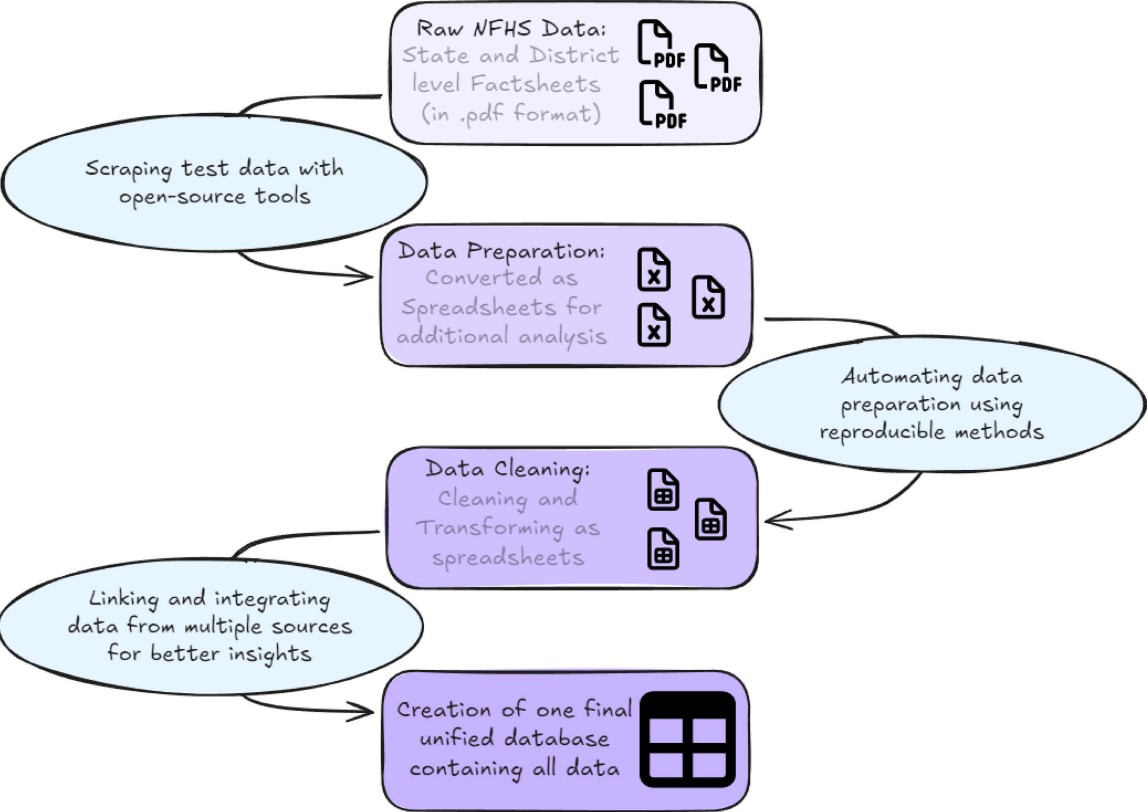

**Fig 1. The Data extraction and preparation Schema.**

## Spatial analysis

Spatial autocorrelation analysis was applied to assess the extent to which districts with similar proportion of C-sections tend to occur next to each other, i.e., form spatial clustering. To examine spatial dependence and clustering of caesarean section deliveries, several spatial techniques were applied, including Global Moran's I, Univariate Local indicator of Spatial Association (LISA), Bivariate Local indicator of Spatial Association (LISA), and LISA cluster mapping. These tests were used to evaluate global spatial autocorrelation, with the null hypothesis asserting that the proportions of C-sections across districts in India are spatially distributed at random [30]. Additionally, to examine High or Low clustering patterns, Getis-Ord General G statistic was used to test for local clustering where, the null hypothesis posits that no spatial clustering of the C-sections exists [31,32]. All spatial analysis was performed using R version 4.4.2 [22].

## Four types of spatial autocorrelation were generated

1. **Hot Spots:** Locations with high values, with similar neighboring values (High-High).
2. **Cold Spots:** Locations with low values, with similar neighboring values (Low-Low).
3. **Spatial Outliers:** Locations with high values, but with low-value neighbors (High-Low).
4. **Spatial Outliers:** Locations with low values, but with higher values of neighbors (Low-High).

All code and data, including the spatial files are available at a public repository (https://github.com/drarunmitra/nfhs-caesarean-paper-data-analysis) along with the computational workflow in R [33].

## Results

### Overall

The proportion of births by cesarean section in India increased from 17.2% in NFHS-4 (2015–16) to 21.5% in NFHS-5 (2019–21). Three zones consistently exceeded the national average in both NFHS-4 and NFHS-5. The South zone exhibited the highest mean proportion of caesarean section births, with 35.38% in NFHS-4 and 39.49% in NFHS-5, while the East zone had the lowest, with 9.97% in NFHS-4 and 14.7% in NFHS-5 across both surveys. The mean increase in c-section births across surveys varied, ranging from 3.71% for the North-East zone to 6.44% in the North zone (See Table 1). At NFHS-5, 21 of 36 states/UTs exceeded the national average. Telangana recorded the highest rates in both surveys (NFHS-4: 57.7%; NFHS-5: 60.7%), while Nagaland had the lowest (NFHS-4: 5.2%; NFHS-5: 5.8%). Punjab showed the largest increase (13.9 percentage points), and Lakshadweep the largest decrease (-7.1 percentage points).Fig 2 illustrates that districts with above-national-average C-section rates were concentrated in Southern India (Andhra Pradesh,

**Table 1. Mean proportion of c-section births by place of residence, place of birth in the administrative zones of India according to NFHS surveys.**

| Zone | NFHS - 4 | | | | | NFHS - 5 | | | | | Increase in CS | | | | |
|---|---|---|---|---|---|---|---|---|---|---|---|---|---|---|---|
| | Overall | Rural | Urban | Public | Private | Overall | Rural | Urban | Public | Private | Overall | Rural | Urban | Public | Private |
| Central | 10.25 | 7.40 | 19.05 | 5.23 | 36.34 | 15.35 | 11.95 | 26.82 | 7.58 | 45.56 | 5.10 | 4.55 | 7.78 | 2.35 | 9.23 |
| East | 9.97 | 8.17 | 20.13 | 5.85 | 39.60 | 14.70 | 12.83 | 25.20 | 8.44 | 49.43 | 4.73 | 4.67 | 5.07 | 2.59 | 9.82 |
| North | 20.57 | 18.05 | 26.96 | 15.78 | 35.77 | 28.34 | 26.51 | 33.21 | 20.30 | 40.53 | 6.44 | 7.05 | 4.19 | 4.52 | 4.76 |
| North East | 13.86 | 9.51 | 27.05 | 12.28 | 44.50 | 17.57 | 13.71 | 29.14 | 14.74 | 52.71 | 3.71 | 4.20 | 2.09 | 2.46 | 8.21 |
| South | 35.38 | 34.96 | 38.59 | 26.76 | 51.09 | 39.49 | 36.50 | 43.38 | 35.65 | 62.98 | 4.11 | 1.54 | 4.79 | 8.89 | 11.89 |
| West | 21.94 | 17.40 | 29.00 | 12.86 | 38.41 | 28.28 | 24.32 | 34.76 | 16.91 | 44.54 | 6.34 | 6.92 | 5.76 | 4.05 | 6.13 |

## Proportion of c−section births (%) among districts of India by NFHS rounds

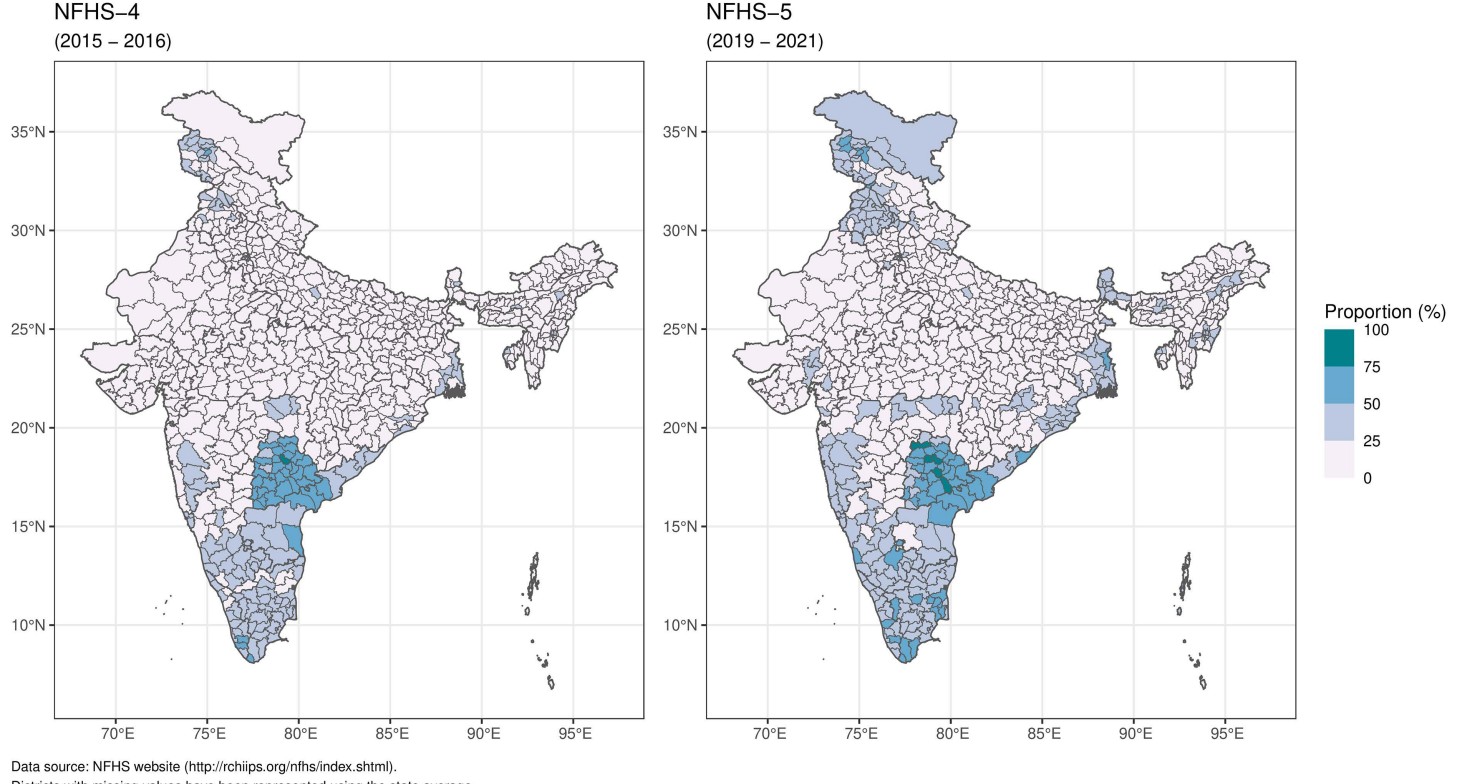

Data source: NFHS website (http://rchiips.org/nfhs/index.shtml).
Districts with missing values have been represented using the state average.

Base map: Administrative boundaries of India from Survey of India (https://www.surveyofindia.gov.in/).

**Fig 2. The spatial distribution of c-section births (%) across different districts in India according to NFHS surveys.**

Kerala, Telangana, Karnataka, Tamil Nadu, Puducherry), Northern India (Punjab, Jammu & Kashmir, NCT Delhi, and Ladakh in NFHS-5), with pockets in Western (Goa, Maharashtra) and Eastern India (West Bengal).

### Urban vs rural place of residence

The increase in C-section rates was more pronounced in rural areas (4.8% increase: from 12.8% to 17.6%) compared to urban areas (4.1% increase: from 28.2% to 32.3%)(Fig A in S1 Text).The South zone maintained the highest mean rural C-section rates (NFHS-4: 35.0%; NFHS-5: 36.5%), while the Central zone had the lowest (NFHS-4: 7.4%; NFHS-5: 12.0%) (Table 1). At NFHS-5, 21 states/UTs exceeded the national rural average of 17.6%. The state-level range extended from 3.6% in rural Nagaland to 58.4% in rural Telangana. Punjab recorded the highest rural increase (14.7 percentage points). The South zone also led urban C-section rates (NFHS-4: 38.6%; NFHS-5: 43.4%). At NFHS-5, 17 states/UTs surpassed the national urban average of 32.3%. Urban rates ranged from 9.8% (Nagaland) to 64.3% (Telangana). The state of Sikkim showed the greatest urban increase of 14.3 percentage points (See Table 2).

### Public vs private health facilities

The proportion of caesarean section births in public health facilities increased by 2.4%, rising from 11.9% in NFHS-4 to 14.3% in NFHS-5. Four zones consistently reported higher caesarean section birth rates compared to the national

**Table 2. Proportion of c-section births by place of residence in states/UTs according to NFHS surveys.**

| | NFHS - 4 | | | NFHS - 5 | | | Increase in CS | | |
|---|---|---|---|---|---|---|---|---|---|
| | Overall | Rural | Urban | Overall | Rural | Urban | Overall | Rural | Urban |
| *South Zone* | | | | | | | | | |
| Andaman & Nicobar Islands | 19.30 | 13.80 | 27.40 | 29.90 | 20.20 | 40.80 | 10.60 | 6.40 | 13.40 |
| Andhra Pradesh | 40.10 | 37.10 | 48.40 | 42.40 | 39.30 | 50.50 | 2.30 | 2.20 | 2.10 |
| Karnataka | 23.60 | 19.80 | 29.10 | 31.50 | 29.40 | 35.20 | 7.90 | 9.60 | 6.10 |
| Kerala | 35.80 | 34.60 | 37.10 | 38.90 | 38.70 | 39.10 | 3.10 | 4.10 | 2.00 |
| Lakshadweep | 38.40 | 49.20 | 36.40 | 31.30 | 33.20 | 30.70 | -7.10 | -16.00 | -5.70 |
| Puducherry | 34.00 | 39.80 | 31.40 | 36.30 | 29.90 | 38.90 | 2.30 | -9.90 | 7.50 |
| Tamil Nadu | 34.10 | 32.30 | 36.10 | 44.90 | 42.90 | 47.50 | 10.80 | 10.60 | 11.40 |
| Telangana | 57.70 | 53.10 | 62.80 | 60.70 | 58.40 | 64.30 | 3.00 | 5.30 | 1.50 |
| *North East Zone* | | | | | | | | | |
| Arunachal Pradesh | 8.90 | 5.80 | 20.10 | 14.80 | 14.40 | 17.10 | 5.90 | 8.60 | -3.00 |
| Assam | 13.40 | 10.80 | 36.80 | 18.10 | 15.60 | 39.20 | 4.70 | 4.80 | 2.40 |
| Manipur | 21.10 | 15.20 | 33.00 | 25.60 | 19.70 | 38.00 | 4.50 | 4.50 | 5.00 |
| Meghalaya | 7.60 | 5.60 | 20.50 | 8.20 | 6.10 | 21.60 | 0.60 | 0.50 | 1.10 |
| Mizoram | 12.70 | 6.00 | 19.00 | 10.80 | 4.80 | 16.80 | -1.90 | -1.20 | -2.20 |
| Nagaland | 5.80 | 3.40 | 12.40 | 5.20 | 3.60 | 9.80 | -0.60 | 0.20 | -2.60 |
| Sikkim | 20.90 | 17.10 | 28.80 | 32.80 | 26.90 | 43.10 | 11.90 | 9.80 | 14.30 |
| Tripura | 20.50 | 12.20 | 45.80 | 25.10 | 18.60 | 47.50 | 4.60 | 6.40 | 1.70 |
| *East Zone* | | | | | | | | | |
| Bihar | 6.20 | 5.40 | 13.90 | 9.70 | 8.80 | 15.70 | 3.50 | 3.40 | 1.80 |
| Jharkhand | 9.90 | 7.00 | 22.40 | 12.80 | 10.20 | 25.80 | 2.90 | 3.20 | 3.40 |
| Odisha | 13.80 | 12.10 | 24.10 | 21.60 | 19.50 | 34.10 | 7.80 | 7.40 | 10.00 |
| *North Zone* | | | | | | | | | |
| Chandigarh | 22.60 | ---- | 23.50 | 31.30 | ---- | 31.70 | 8.70 | ---- | 8.20 |
| Haryana | 11.70 | 10.60 | 13.60 | 19.50 | 17.80 | 23.50 | 7.80 | 7.20 | 9.90 |
| Himachal Pradesh | 16.70 | 15.60 | 29.60 | 21.00 | 20.30 | 26.20 | 4.30 | 4.70 | -3.40 |
| Jammu & Kashmir | 33.10 | 26.90 | 53.10 | 41.70 | 37.80 | 54.70 | 8.60 | 10.90 | 1.60 |
| Ladakh | ---- | ---- | ---- | 37.60 | 35.00 | 47.70 | ---- | ---- | ---- |
| NCT Delhi | 26.70 | 25.00 | 26.70 | 26.70 | 28.20 | 23.40 | 0.00 | 3.20 | -3.30 |
| Punjab | 24.60 | 23.70 | 25.80 | 38.50 | 38.40 | 38.80 | 13.90 | 14.70 | 13.00 |
| Rajasthan | 8.60 | 6.50 | 16.40 | 10.40 | 8.10 | 19.70 | 1.80 | 1.60 | 3.30 |
| *Central Zone* | | | | | | | | | |
| Chhattisgarh | 9.90 | 7.50 | 18.90 | 15.20 | 11.30 | 31.20 | 5.30 | 3.80 | 12.30 |
| Madhya Pradesh | 8.60 | 5.00 | 19.10 | 12.10 | 8.80 | 23.30 | 3.50 | 3.80 | 4.20 |
| Uttar Pradesh | 9.40 | 6.90 | 18.90 | 13.70 | 11.00 | 24.20 | 4.30 | 4.10 | 5.30 |
| Uttarakhand | 13.10 | 10.20 | 19.30 | 20.40 | 16.70 | 28.60 | 7.30 | 6.50 | 9.30 |
| *West Zone* | | | | | | | | | |
| DNH & DD | 16.00 | 13.20 | 20.80 | 22.90 | 16.10 | 29.90 | 6.90 | 2.90 | 9.10 |
| Goa | 31.40 | 27.70 | 33.50 | 39.50 | 40.10 | 39.10 | 8.10 | 12.40 | 5.60 |
| Gujarat | 18.40 | 12.00 | 27.80 | 21.00 | 15.30 | 30.70 | 2.60 | 3.30 | 2.90 |
| Maharashtra | 20.10 | 15.20 | 26.30 | 25.40 | 21.50 | 30.60 | 5.30 | 6.30 | 4.30 |
| West Bengal | 23.80 | 18.90 | 36.60 | 32.60 | 28.60 | 43.50 | 8.80 | 9.70 | 6.90 |

average for public health facilities in both NFHS-4 and NFHS-5 (See Table 1). The South zone had the highest rates (NFHS-4: 26.76%; NFHS-5: 35.65%) and the Central zone the lowest (NFHS-4: 5.23%; NFHS-5: 7.58%). At NFHS-5, 24 states/UTs exceeded the national public facility average. State-level rates ranged from 3.6% (Bihar) to 44.5% (Telangana), with Sikkim showing the largest increase (12.3 percentage points) (Table 3). Districts exceeding national averages were concentrated in Southern India (Kerala, Puducherry, Telangana, Tamil Nadu, Andhra Pradesh, Karnataka), Northern India (Jammu & Kashmir, Punjab), North-Eastern India (Sikkim, Manipur, Assam), and Eastern India (West Bengal, Odisha) (See Fig 3).

Private facilities showed a larger increase of 6.5 percentage points (from 40.9% to 47.4%). The South zone again led (NFHS-4: 51.09%; NFHS-5: 62.98%), while the North zone had the lowest rates (NFHS-4: 35.77%; NFHS-5: 40.53%). At NFHS-5, 13 states/UTs exceeded the national private facility average. State-level rates ranged from 23.6% (Nagaland) to 82.7% (West Bengal), with Assam showing the greatest increase with 17.3 percentage points (See Table 3). High-rate districts were predominantly located along the eastern coast (Andhra Pradesh, Odisha, West Bengal), in Southern India (Telangana, Tamil Nadu, Karnataka), North-Eastern India (Assam, Tripura, Sikkim), and Northern India (Jammu & Kashmir) (See Fig 3).

The median proportion of C-sections increased the most in private health facilities in rural areas (8.0%), followed by public health facilities in rural areas (5.8%), private health facilities in urban areas (4.8%), and public health facilities in urban areas (2.8%) between NFHS-4 and NFHS-5 (Fig B in S1 Text). Further analysis using Analysis of Variance (ANOVA) indicated that the effects of place of residence (p < 0.001), NFHS survey (p = 0.007), and place of birth (p < 0.001) on the proportion of caesarean section births were statistically significant (Table A in S1 Text).

## Spatial clustering & hotspot analysis

Further analysis to assess clustering in districts with high caesarean section rates revealed highly significant global spatial autocorrelation. Global Moran's I (p < 0.01) and Getis-Ord General G statistics (p < 0.01) confirmed significant spatial autocorrelation and clustering for both survey rounds and facility types (Table B in S1 Text).

In NFHS-4, high caesarean section rates (hot spots) were predominantly observed in districts of Southern India (Telangana, Andhra Pradesh, Karnataka, Kerala, Tamil Nadu) with additional pockets in Northern India (Jammu & Kashmir) and Eastern India (West Bengal). Cold spots were primarily found in districts of Central India (Madhya Pradesh, Chhattisgarh, Uttar Pradesh, Uttarakhand), Eastern India (Bihar, Jharkhand, Odisha), and North-Eastern India (Arunachal Pradesh, Assam, Nagaland, Meghalaya, Mizoram), with localized pockets in Western India (Gujarat, Maharashtra) and Northern India (Rajasthan). In NFHS-5, hot spots were again predominantly located in Southern India (Telangana, Andhra Pradesh, Karnataka, Kerala, Tamil Nadu, Puducherry) and Northern India (Jammu & Kashmir, Ladakh, Punjab), with a pocket in Eastern India (West Bengal). Cold spots remained concentrated in Central India (Madhya Pradesh, Chhattisgarh, Uttar Pradesh, Uttarakhand), Eastern India (Bihar, Jharkhand, Odisha), and North-Eastern India (Arunachal Pradesh, Assam, Nagaland, Meghalaya, Mizoram), with additional pockets in Western India (Gujarat, Maharashtra) and Northern India (Rajasthan) (See Fig 4).

As illustrated in Fig 5, the spatial distribution of caesarean section clustering (high/low) varies between public and private health facilities. In NFHS-4, high caesarean section rates (hot spots) in public health facilities were predominantly observed in districts of southern India (Telangana, Andhra Pradesh, Karnataka, Kerala and Tamil Nadu) with additional pockets in Northern (Jammu & Kashmir) and Eastern India (West Bengal). Cold spots were primarily found in Central India (Madhya Pradesh, Chhattisgarh, Uttar Pradesh), Eastern India (Bihar and Jharkhand) with pockets in Western India (Gujarat and Maharashtra) and Northern India (Rajasthan and Haryana).In NFHS-5, hot spots for public health facilities remained concentrated in districts of Southern India (Telangana, Andhra Pradesh, Karnataka, Kerala and Tamil Nadu) with additional pockets in Northern India (Jammu & Kashmir and Punjab) and Eastern India (West Bengal). Cold spots continued to be present in the districts of Central India (Madhya Pradesh, Chhattisgarh, Uttar Pradesh), Eastern India (Bihar and Jharkhand) with pockets in Western India (Gujarat and Maharashtra) and Northern India (Rajasthan)

**Table 3. Proportion of c-section births by place of birth in states/UTs according to NFHS surveys.**

| State | NFHS - 4 | | | NFHS - 5 | | | Increase in CS | | |
|---|---|---|---|---|---|---|---|---|---|
| | Overall | Public | Private | Overall | Public | Private | Overall | Public | Private |
| *South* | | | | | | | | | |
| Andaman & Nicobar Islands | 19.30 | 16.90 | ---- | 29.90 | 23.60 | ---- | 10.60 | 6.70 | ---- |
| Andhra Pradesh | 40.10 | 25.50 | 57.00 | 42.40 | 26.60 | 63.00 | 2.30 | 1.10 | 6.00 |
| Karnataka | 23.60 | 16.90 | 40.30 | 31.50 | 22.60 | 52.50 | 7.90 | 5.70 | 12.20 |
| Kerala | 35.80 | 31.40 | 38.60 | 38.90 | 37.20 | 39.90 | 3.10 | 5.80 | 1.30 |
| Puducherry | 34.00 | 31.00 | 47.40 | 36.30 | 34.80 | 42.00 | 2.30 | 3.80 | -5.40 |
| Tamil Nadu | 34.10 | 26.30 | 51.30 | 44.90 | 36.00 | 63.80 | 10.80 | 9.70 | 12.50 |
| Telangana | 57.70 | 40.30 | 74.50 | 60.70 | 44.50 | 81.50 | 3.00 | 4.20 | 7.00 |
| Lakshadweep | 38.40 | 27.10 | 59.90 | 31.30 | 28.20 | 37.70 | -7.10 | 1.10 | -22.20 |
| *North East* | | | | | | | | | |
| Arunachal Pradesh | 8.90 | 12.50 | 37.50 | 14.80 | 17.00 | 47.30 | 5.90 | 4.50 | 9.80 |
| Assam | 13.40 | 12.90 | 53.30 | 18.10 | 15.20 | 70.60 | 4.70 | 2.30 | 17.30 |
| Manipur | 21.10 | 22.60 | 46.20 | 25.60 | 24.70 | 53.20 | 4.50 | 2.10 | 7.00 |
| Meghalaya | 7.60 | 9.80 | 31.40 | 8.20 | 9.20 | 40.80 | 0.60 | -0.60 | 9.40 |
| Mizoram | 12.70 | 12.30 | 30.10 | 10.80 | 9.80 | 30.40 | -1.90 | -2.50 | 0.30 |
| Nagaland | 33.60 | 13.40 | 31.50 | 5.20 | 8.00 | 23.60 | -28.40 | -5.40 | -7.90 |
| Sikkim | 20.90 | 18.10 | 49.30 | 32.80 | 30.40 | 55.40 | 11.90 | 12.30 | 6.10 |
| Tripura | 20.50 | 18.10 | 73.70 | 25.10 | 22.70 | 69.30 | 4.60 | 4.60 | -4.40 |
| *East* | | | | | | | | | |
| Bihar | 6.20 | 2.60 | 31.00 | 9.70 | 3.60 | 39.60 | 3.50 | 1.00 | 8.60 |
| Jharkhand | 9.90 | 4.60 | 39.50 | 12.80 | 7.00 | 46.70 | 2.90 | 2.40 | 7.20 |
| Odisha | 13.80 | 11.50 | 53.70 | 21.60 | 15.30 | 70.70 | 7.80 | 3.80 | 17.00 |
| *North* | | | | | | | | | |
| Chandigarh | 22.60 | 19.50 | 44.00 | 31.30 | 30.40 | ---- | 8.70 | 10.90 | ---- |
| Haryana | 11.70 | 8.60 | 25.30 | 19.50 | 11.70 | 33.90 | 7.80 | 3.10 | 8.60 |
| Himachal Pradesh | 16.70 | 16.40 | 44.40 | 21.00 | 17.40 | 51.40 | 4.30 | 1.00 | 7.00 |
| Jammu & Kashmir | 33.10 | 35.10 | 75.50 | 41.70 | 42.70 | 82.10 | 8.60 | 7.60 | 6.60 |
| Ladakh | ---- | ---- | ---- | 37.60 | 39.30 | ---- | ---- | ---- | ---- |
| Punjab | 24.60 | 17.80 | 39.70 | 38.50 | 29.90 | 55.50 | 13.90 | 12.10 | 15.80 |
| Rajasthan | 8.60 | 6.10 | 23.20 | 10.40 | 7.20 | 26.90 | 1.80 | 1.10 | 3.70 |
| NCT Delhi | 26.70 | 26.50 | 41.50 | 26.70 | 26.50 | 41.50 | 0.00 | 0.00 | 0.00 |
| *Central* | | | | | | | | | |
| Chhattisgarh | 9.90 | 5.70 | 46.60 | 15.20 | 8.90 | 57.00 | 5.30 | 3.20 | 10.40 |
| Madhya Pradesh | 8.60 | 5.80 | 40.80 | 12.10 | 8.20 | 52.30 | 3.50 | 2.40 | 11.50 |
| Uttarakhand | 13.10 | 9.30 | 36.40 | 20.40 | 14.00 | 43.30 | 7.30 | 4.70 | 6.90 |
| Uttar Pradesh | 9.40 | 4.70 | 31.30 | 13.70 | 6.20 | 39.40 | 4.30 | 1.50 | 8.10 |
| *West* | | | | | | | | | |
| DNH & DD | 16.00 | 9.65 | 32.60 | 22.90 | 17.30 | 42.50 | 6.90 | 7.65 | 9.90 |
| DNH & DD | 16.00 | 11.10 | 33.20 | 22.90 | 17.30 | 42.50 | 6.90 | 6.20 | 9.30 |
| Goa | 31.40 | 19.90 | 51.30 | 39.50 | 31.50 | 50.00 | 8.10 | 11.60 | -1.30 |
| Gujarat | 18.40 | 10.80 | 26.60 | 21.00 | 12.40 | 30.80 | 2.60 | 1.60 | 4.20 |
| Maharashtra | 20.10 | 13.10 | 33.10 | 25.40 | 18.30 | 39.10 | 5.30 | 5.20 | 6.00 |
| West Bengal | 23.80 | 18.80 | 70.90 | 32.60 | 22.90 | 82.70 | 8.80 | 4.10 | 11.80 |

## Proportion of c–section births (%) among districts of India by type of facility

### (A) Public Health Facilities

NFHS–4
(2015 – 2016)

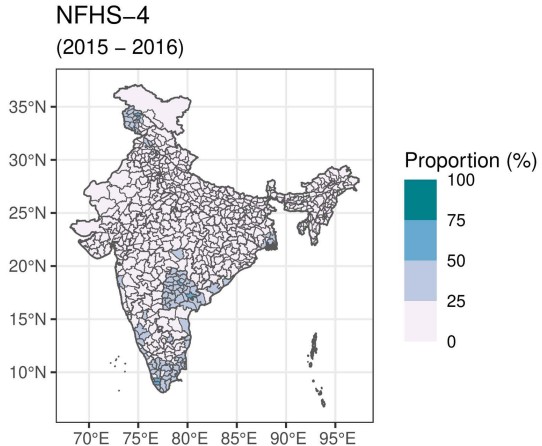

NFHS–5
(2019 – 2021)

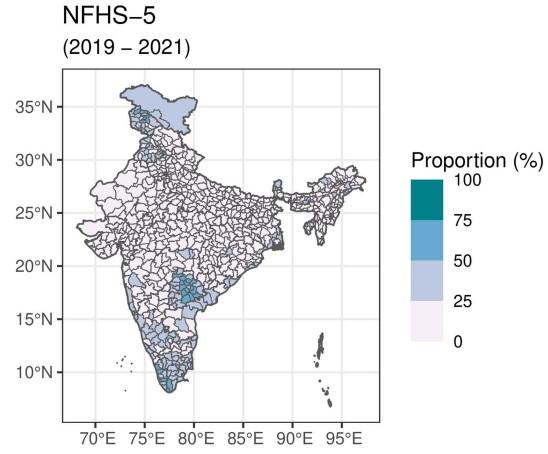

### (B) Private Health Facilities

NFHS–4
(2015 – 2016)

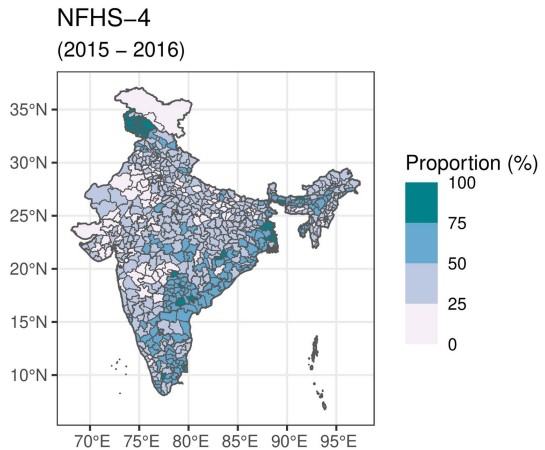

NFHS–5
(2019 – 2021)

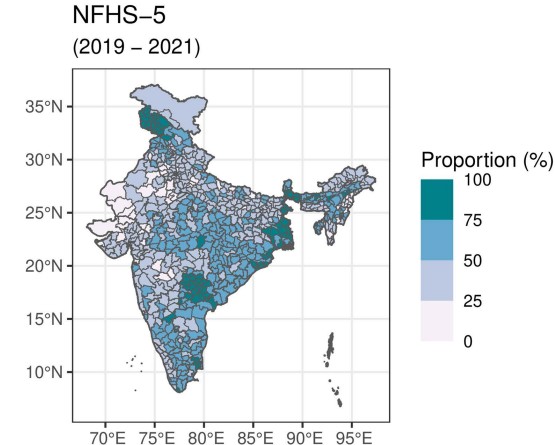

Data source: NFHS website (http://rchiips.org/nfhs/index.shtml).
Districts with missing values have been represented using the state average.

Base map: Administrative boundaries of India from Survey of India (https://www.surveyofindia.gov.in/).

**Fig 3. Proportion of C-Section births across districts in India, comparing (A) public health facilities and (B) private health facilities, for NFHS-4 (2015–2016) and NFHS-5 (2019–2021).**

In NFHS-4, the clustering of high caesarean section rates (hot spots) in private health facilities was observed predominantly in the districts of Southern India (Telangana, Andhra Pradesh, Tamil Nadu). Additional hot spots were identified in Eastern India (West Bengal, Odisha), with isolated occurrences in Northern India (Jammu & Kashmir) and North-Eastern India (Assam, Sikkim). Cold spots, conversely, were concentrated in the districts of Western India (Gujarat, Maharashtra) and Northern India (Rajasthan, Haryana), with additional pockets found in Central India (Uttar Pradesh), Eastern India

## Clustering of C–Section Births (%) across districts in India by NFHS rounds

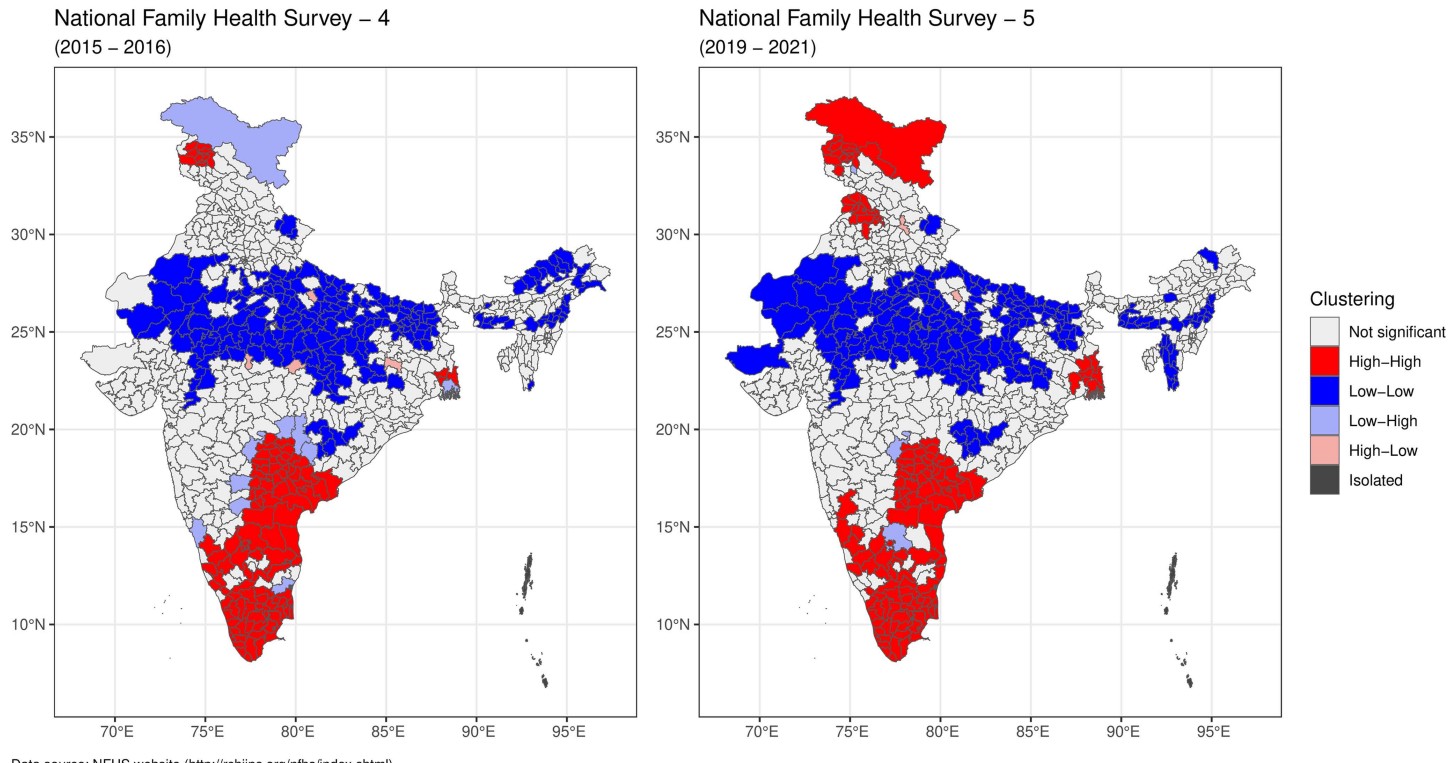

National Family Health Survey – 4
(2015 – 2016)

National Family Health Survey – 5
(2019 – 2021)

Clustering
- Not significant
- High–High
- Low–Low
- Low–High
- High–Low
- Isolated

Data source: NFHS website (http://rchiips.org/nfhs/index.shtml).
Districts with missing values have been represented using the state average.

Base map: Administrative boundaries of India from Survey of India (https://www.surveyofindia.gov.in/).

**Fig 4. Clustering of C-Section Births (%) across districts in India, comparing NFHS-4 (2015–2016) and NFHS-5 (2019–2021). Spatial clusters identified using Local Moran's I analysis with queen contiguity weights. Cluster types: High-High (high C-section districts surrounded by high neighbors), Low-Low (low districts surrounded by low neighbors), High-Low and Low-High (spatial outliers), and Not Significant (no significant spatial autocorrelation).**

(Bihar), and Southern India (Karnataka).). In NFHS-5, the hot spots for caesarean section births in private health facilities were primarily concentrated along the eastern coast, notably in Andhra Pradesh, Odisha, and West Bengal, with additional pockets in North-Eastern India (Assam, Tripura, Sikkim), Southern India (Telangana, Tamil Nadu, Karnataka), and Northern India (Jammu & Kashmir). Cold spots for private health facilities were identified in Western India (Maharashtra, Gujarat), Northern India (Rajasthan, Haryana), with additional pockets in Central India (Uttar Pradesh), Eastern India (Bihar), and North-Eastern India (Meghalaya, Nagaland, Mizoram).

## Discussion

Our study found that the C-section rate in India has steadily increased from 17.2% to 21.5%, well exceeding the WHO's recommendation, with significant variations observed across different states and districts. The rise in C-section rates is a global phenomenon, with rates surpassing the WHO recommended threshold in several countries [34,35]. In South Asia, where India is located, the C-section rate has more than doubled between 2000–2015. A rise in institutional births has been implicated as the major driver for this rising trend, contributing to 66.5% of the increase in the global C-section rate and 95.4% of the increase in the C-section rate for South Asia [34]. Variations in C-section rates are also influenced by other factors such as residence, education, wealth [34,36–38]. The high rate of C-sections in urban areas is attributed to

# Clustering of C–Section Births (%) across districts in India by type of facility

## (A) Public Health Facilities

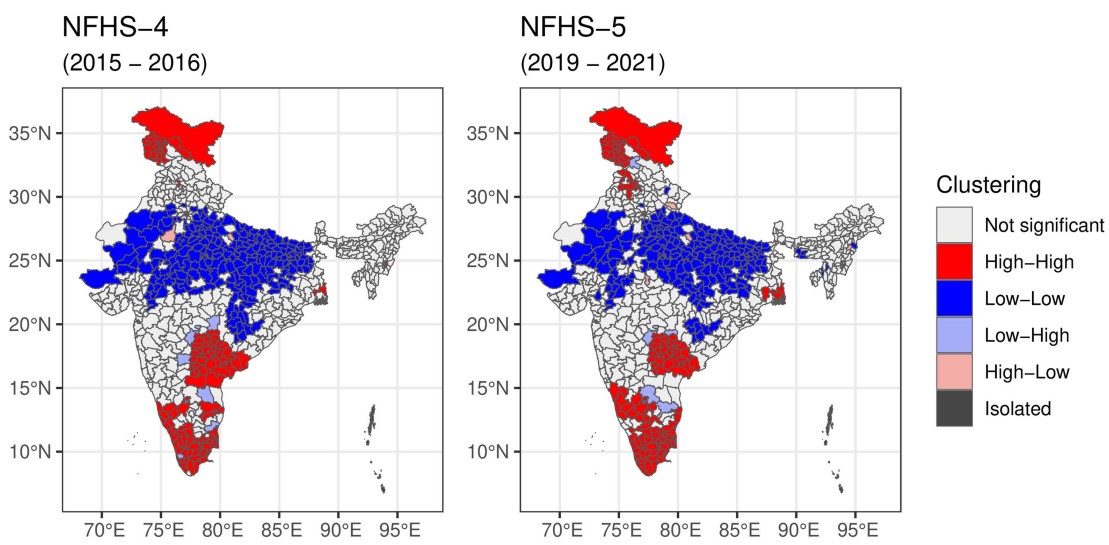

## (B) Private Health Facilities

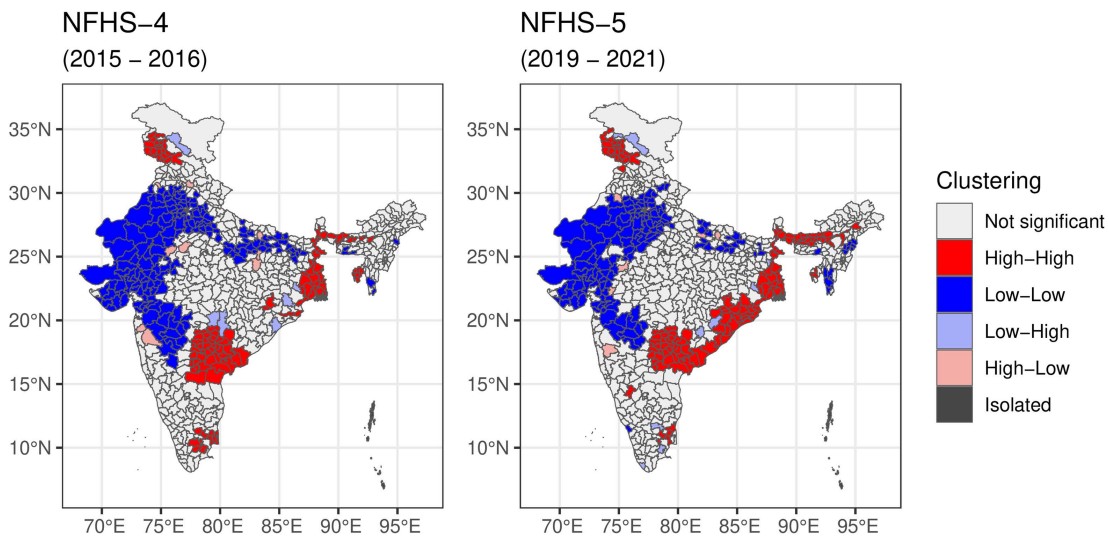

Data source: NFHS website (http://rchiips.org/nfhs/index.shtml).
Districts with missing values have been represented using the state average.

Base map: Administrative boundaries of India from Survey of India (https://www.surveyofindia.gov.in/).

**Fig 5. Clustering of C-Section Births (%) across districts in India, comparing (A) public health facilities and (B) private health facilities, for NFHS-4 and NFHS-5.** Spatial clusters identified using Local Moran's I analysis.

include the availability of advanced healthcare facilities equipped with modern obstetric technologies, a stronger preference for private healthcare services, and the profit-driven nature of the private healthcare sector [39–42]. However, as previously thought, urbanization, wealth and education may no longer be the primary drivers of the increase in C-sections. The results of our study indicate that the increase in C-section rates in no longer a uniquely urban problem. The growth in C-section rates has been more pronounced in rural areas than urban areas. This picture is further compounded by the type of healthcare provider, with the rise in C-section rates being the highest for private healthcare facilities in rural areas. Private healthcare facilities typically have higher C-section rates when compared to public healthcare facilities, a pattern also noted in other studies from Asia, Africa, Latin America, and the Caribbean [36,37,43–46].

The recent entry of private healthcare in rural India coupled with policy initiatives, such as the push for 100% institutional deliveries may have contributed to this situation. The resultant influx of pregnant women to public healthcare facilities, which are particularly resource constrained in rural regions could have resulted in a demand-supply mismatch. This vacuum in service delivery would then have been filled by the private healthcare sector, as an increasing number of pregnant women would have been compelled to avail their services. Several factors play a role in the high prevalence of unnecessary C-sections in private healthcare. Driven by profit, some private providers may perform medically unindicated caesarean deliveries. Some women may request a C-section as they fear of labour pain or misperceive that they are faster and safer. The inadequate training of healthcare providers in vaginal delivery techniques may have also contributed to the rising rates [47–51]. Additionally, factors such as increased awareness, changing healthcare preferences, improved access to healthcare services, and shifts in medical practices have influenced the mode of delivery in rural India [6,52,53]. Further examination of these policy-related issues is warranted to gain a more comprehensive understanding of the root causes

The overuse of C-sections has significant implications; they can burden health systems in a developing country like India and result in adverse maternal, perinatal infant health outcomes, impeding our pursuit of the Sustainable Developmental Goals (SDG) [54]. Moreover, it is estimated that excessive C-sections result in an annual cost of 5130 crore Indian National Rupees (INR) or 599 million US Dollars (USD) to Indian families [55]. Their overuse can therefore lead to financial strain and catastrophic health expenditures (CHE), pushing people into poverty and aggravating existing socioeconomic inequities [18,56]. The implications of a rising C-section rate can therefore be particularly acute in rural India where health systems are resource constrained and populations are generally socioeconomically worse off. Consequently, it is imperative to curb the excessive rise in C-sections, which has become a widespread phenomenon.

Geospatial analytical techniques have been utilised in previous research on maternal and child health [35,57,58]. The difficulties in translating large-scale data into policy are overcome by these techniques, which are leveraged to transform data into valuable and actionable insights. The present analysis provides critical information on regions with high C-sections through clustering or hotspots. Overall, states such as Telangana, Andhra Pradesh, Tamil Nadu, Jammu & Kashmir, West Bengal consistently exhibited hot spots. Hot spots also emerged in Punjab, which had the highest increase in C-sections across NFHS surveys. The hotspots for private facilities were predominantly concentrated in the southern and eastern coastal regions of India with a few in Assam, Jammu & Kashmir and Tripura. For public facilities hotspots were mainly seen in Telangana, Andhra Pradesh, Tamil Nadu, Jammu & Kashmir and West Bengal. Understanding the factors contributing to the rise of C-section births in public health settings is critical, as these facilities are not financially incentivized to perform such procedures.

Certain factors, such as socioeconomic status, education, and healthcare access, have been studied in relation to the rise in C-sections in India [12,13,46]. However, numerous others require investigation to fully understand this multifaceted issue. Therefore, it is crucial to conduct localized research, including qualitative studies to explore the underlying social, cultural, and medical factors contributing to this trend, which will inform interventions to reduce unnecessary C-sections. Some potential solutions include periodic C-section audits to discourage medically unnecessary procedures. Initiatives such as the encouragement of trial of labor after C-section (TOLAC) leading to a successful vaginal birth after C-section

(VBAC) and enhanced training in assisted vaginal deliveries, could also play a pivotal role. Finally, quality improvement initiatives aimed at reducing C-section rates in private healthcare settings must also be explored to mitigate this growing trend [59].

Our findings complement and extend prior spatial analyses of C-sections in India. A study done in South India reported significant regional disparities in C-section utilization, identifying clustering driven by maternal age, education, and healthcare facility type [18]. Similarly, another study identified spatial hotspots of C-section rates in districts of Sikkim, Tripura, Assam, and Manipur using NFHS-5 data for the northeastern region [19]. While these studies provide valuable subregional insights, the present study advances the evidence base by offering a nationwide perspective encompassing all districts across India, examining the temporal evolution of clustering from NFHS-4 to NFHS-5, and critically, disaggregating the analysis by public and private facilities. This facility-level disaggregation reveals that private facility hotspots concentrate predominantly along the eastern coast and in northeastern regions, while public facility hotspots are predominantly in southern India; a distinction not captured by previous studies. These differential patterns underscore that policy interventions to address unnecessary C-sections must be tailored to the distinct dynamics of the public and private healthcare sectors.

## Strengths & limitations

The major strength of this study is that it is to the best of our knowledge, the most comprehensive spatial analysis of C-sections at the district level. Moreover, the use of nationally representative survey data and robust geospatial techniques lend further strength to our findings. However, it is not without its limitations. Geospatial techniques for disaggregated data are computationally intensive due to which aggregate data have been used. This prevented the exploration of individual level factors associated with C-sections. Another limitation is that factors such as total fertility rate, health insurance coverage, and medically indicated repeat C-sections have not been accounted for. Future research should incorporate these variables to provide a more comprehensive understanding of the determinants influencing the rise in C-section rates.

## Conclusion

As demonstrated in this study, a significant disparity exists in the proportion of C-section births between public and private healthcare facilities, further complicated by urban-rural differences. The push for 100% institutional deliveries may have resulted in unintended consequences in underserved regions, where private healthcare facilities address the unmet need for quality maternal care. These insights, derived from robust scientific methodologies, provide valuable opportunities for course correction and inform decision-making regarding public health interventions and health systems strengthening. In conclusion, the study underscores the utility of geospatial methodologies applied to large-scale survey data to uncover inequities in maternal healthcare, with important implications for public health policy.

## Supporting information

**S1 Text. Supplement.** This document contains additional methods, tables (Table A & Table B), and figures (Fig A & Fig B). (DOCX)

## Author contributions

**Conceptualization:** Durgesh Prasad Sahoo, Biju Soman, Neeraj Agarwal, Arun Mitra.

**Data curation:** Biju Soman, Arun Mitra.

**Formal analysis:** Biju Soman, Shreyas Patil, Arun Mitra.

**Methodology:** Durgesh Prasad Sahoo, Biju Soman, Neeraj Agarwal, Shreyas Patil, Arun Mitra.

**Project administration:** Durgesh Prasad Sahoo, Biju Soman, Arun Mitra.

**Resources:** Biju Soman, Arun Mitra.

**Software:** Biju Soman, Shreyas Patil, Arun Mitra.

**Supervision:** Durgesh Prasad Sahoo, Biju Soman, Neeraj Agarwal, Arun Mitra.

**Validation:** Biju Soman, Arun Mitra.

**Visualization:** Biju Soman, Shreyas Patil, Adrija Roy, Arun Mitra.

**Writing – original draft:** Durgesh Prasad Sahoo, Biju Soman, Neeraj Agarwal, Shreyas Patil, Arun Mitra.

**Writing – review & editing:** Durgesh Prasad Sahoo, Biju Soman, Neeraj Agarwal, Shreyas Patil, Adrija Roy, Arun Mitra.

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
