## [Decision Letter · Decision Letter 0]

11 Sep 2025

PGPH-D-25-01044

Exploring Spatial Clusters of Caesarean Sections across India - Insights from National Family Health Survey Data

Dear Dr. Mitra

Thank you for submitting your manuscript to PLOS Global Public Health. After careful consideration, we feel that it has merit but does not fully meet PLOS Global Public Health’s publication criteria as it currently stands. Therefore, we invite you to submit a revised version of the manuscript that addresses the points raised during the review process.

Your manuscript has been assessed by two reviewers. While they show interest in your work, they have raised numerous comments which require your attention. Please consider these points carefully when preparing your revised manuscript and point-by-point response document.

We look forward to receiving your revised manuscript.

Kind regards

Dr Jason Morgan

Staff Editor

Journal Requirements:

1. In the online submission form, you indicated that The data used in the current study are available in the public domain. However, the spatial datafiles and the algorithms for cleaning, organizing and analysis would be made available from the corresponding.

3. Uploaded as supplementary information.

2. Please provide separate figure files in .tif or .eps format.

4. We notice that your supplementary figures are uploaded with the file type 'Figure'. Please amend the file type to 'Supporting Information'. Please ensure that each Supporting Information file has a legend listed in the manuscript after the references list.

Reviewers' comments:

Reviewer's Responses to Questions

**Comments to the Author**

1. Does this manuscript meet PLOS Global Public Health’s publication criteria?

Reviewer #1: Partly

Reviewer #2: Yes

2. Has the statistical analysis been performed appropriately and rigorously?

Reviewer #1: Yes

Reviewer #2: Yes

3. Have the authors made all data underlying the findings in their manuscript fully available (please refer to the Data Availability Statement at the start of the manuscript PDF file)?

Reviewer #1: Yes

Reviewer #2: Yes

4. Is the manuscript presented in an intelligible fashion and written in standard English?

Reviewer #1: Yes

Reviewer #2: Yes

Reviewer #1: Minor Errors and typos

line 16 clustering-----clusters

line 17 Global Moran's---- add index/statistic

line 35 delivery compared ----- delivery as compared

line 39 TThe ---- The

line 47 focus of research ---- focus of many researchers

line 57 put space after full stop in (6).In

line 98 R(?)

line 159-161 Three zones ---- NFHS 5. repeated twice

line 165 delete (6.44%) duplicated.

Major Correction

1 In line 159, You have mentioned that the national average C-section. Could please specify the source and does it contradict with your assumption that it is a first kind of research in India.

2. It would be better, if you could have made the description of the descriptive statistics short and precise instead of making duplicated similar results.

3. In figure 4, at the northern tip region, the graph shows a change from Low-High to High-High spatial distribution. But when you come to figure five there is no change in both private and public hospitals. could you please figure it out.

4. In figure 5, figures a and b are not labeled clearly.

5 Tables S1 and S2 are the main results of the research but not yet discussed and concluded in detail. Hence, I would suggest including them in the main results and discuss in detail.

6 could you please show where is the result of the analysis of Moren's I index/statistic.

7 My final recommendation would be most of the contents of the discussion are literature reviews. It would be better I it is cut and paste in the introduction section.

Reviewer #2: The manuscript presents original findings derived from NFHS-4 and NFHS-5 data. While previous studies have documented caesarean section trends in India, this work provides a novel perspective by applying spatial clustering techniques at both state and district levels. This geospatial approach enhances the originality and policy relevance of the study.

There is no indication that the manuscript, in whole or in part, has been previously published. The work appears to present new analyses based on publicly available datasets.

The statistical analysis is explained with sufficient detail. The authors clearly describe the use of ANOVA for testing differences in proportions, and advanced spatial techniques including Global Moran’s I, Getis-Ord General G, and univariate/bivariate LISA cluster mapping. The workflow of data extraction, cleaning, and preparation is well documented, and reproducibility is facilitated by the explicit mention of R packages and versions.

The conclusions are consistent with and supported by the results. The manuscript demonstrates that caesarean section rates have increased significantly in India, with particularly pronounced rises in rural areas and in private health facilities. These conclusions are backed by descriptive statistics, tables, maps, and spatial clustering results

The manuscript is written in comprehensible English. The scientific meaning is clear throughout, although minor grammatical and stylistic editing could improve fluency.

The study complies with ethical standards. Ethical approval was obtained from the Institutional Ethics Committee (AIIMS/BBN/IEC/AUG/2024). Data were sourced from nationally available NFHS surveys, which are publicly accessible. No ethical concerns are identified.

The authors state that NFHS-4 and NFHS-5 datasets are publicly available. In addition, the spatial data files and analytic algorithms will be made available from the corresponding author upon request. This complies with the journal’s data availability policy.

I consider the article suitable for publication after minor revisions focused on improving the fluency of English writing and expanding the discussion of actionable policy implications.

**Do you want your identity to be public for this peer review?** For information about this choice, including consent withdrawal, please see our Privacy Policy

Reviewer #1: No

Reviewer #2: **Yes:** Ramon Escuriet

---

## [Decision Letter · Decision Letter 1]

2 Feb 2026

PGPH-D-25-01044R1

Exploring Spatial Clusters of Caesarean Sections across India - Insights from National Family Health Survey Data

Dear Dr. Mitra,

Thank you for submitting your manuscript to PLOS Global Public Health. After careful consideration, we feel that it has merit but does not fully meet PLOS Global Public Health’s publication criteria as it currently stands. Therefore, we invite you to submit a revised version of the manuscript that addresses the points raised during the review process.

The reviewer is positive about the study. Before we can proceed to Acceptance, please address the following:

Previous studies have investigated the spatial clustering of Caesarean sections in India, e.g. https://doi.org/10.1007/s13224-025-02150-4 and https://doi.org/10.1186/s12889-024-19984-8. This is relevant literature that should be cited. Additionally, please discuss the contribution of the current study in the context of this previously published research.

We look forward to receiving your revised manuscript.

Kind regards,

Alejandro Torrado Pacheco, PhD

Staff Editor

Journal Requirements:

Additional Editor Comments (if provided):

Reviewers' comments:

Reviewer's Responses to Questions

**Comments to the Author**

Reviewer #1: All comments have been addressed

publication criteria?

Reviewer #1: Yes

3. Has the statistical analysis been performed appropriately and rigorously?

Reviewer #1: Yes

4. Have the authors made all data underlying the findings in their manuscript fully available (please refer to the Data Availability Statement at the start of the manuscript PDF file)?

Reviewer #1: Yes

5. Is the manuscript presented in an intelligible fashion and written in standard English?

Reviewer #1: Yes

Reviewer #1: (No Response)

**Do you want your identity to be public for this peer review?** For information about this choice, including consent withdrawal, please see our Privacy Policy

Reviewer #1: No

---

## [Editor Report · Decision Letter 2]

12 Feb 2026

Exploring Spatial Clusters of Caesarean Sections across India - Insights from National Family Health Survey Data

PGPH-D-25-01044R2

Dear Mitra,

We are pleased to inform you that your manuscript 'Exploring Spatial Clusters of Caesarean Sections across India - Insights from National Family Health Survey Data' has been provisionally accepted for publication in PLOS Global Public Health.

Best regards,

Julia Robinson

Executive Editor